# Detection and Recognition of Visual Geons Based on Specific Object-of-Interest Imaging Technology

**DOI:** 10.3390/s25103022

**Published:** 2025-05-10

**Authors:** Yonghao Wu, Minyi Liu, Jun Li

**Affiliations:** School of Electronic Science and Engineering (School of Microelectronics), South China Normal University, Foshan 528000, China

**Keywords:** object-of-interest imaging, visual geons, visual attention, deep learning

## Abstract

Across domains such as visual processing, computer graphics, neuroscience, and biological sciences, geons are recognized as fundamental components of complex shapes. Their theoretical significance has been extensively acknowledged in scientific research. However, accurately identifying and extracting these structural components remains a persistent challenge. This study integrates theoretical foundations from signal processing, computer graphics, neuroscience, and biological sciences. We employ specific object-of-interest imaging and neural networks to mathematically operationalize visual geon characterization, thereby elucidating their intrinsic properties. Experiments validate the core hypothesis of geon theory, namely that geons are foundational components for the visual system to recognize complex objects. Through training, neural networks are capable of identifying distinct basic geons and, on this basis, performing target recognition in more complex scenarios. These findings provide empirical confirmation of geons’ existence and their critical role in visual recognition, establishing novel computational paradigms and theoretical foundations for interdisciplinary research.

## 1. Introduction

The generation and interpretation of complex geometric structures follow hierarchical construction principles, in which a finite set of core topological elements is combined to form higher-order visual representations through specific combinatorial rules. Geons are widely recognized by researchers as fundamental units underlying all visual–graphic representations, also referred to as geometric primitives or simply primitives. Since the establishment of their theoretical foundation in Marr’s computational vision framework, formal representations of geons have progressed significantly in computer graphics, cognitive neuroscience, and biological sciences, highlighting the pivotal role of geon theory in explainable artificial intelligence and bioinspired computing.

### 1.1. Recognition-by-Components (RBCs) Framework Based on Marr

In the 1980s, Biederman et al. proposed the RBCs theory [1,2]. This theory systematically explains the conceptual framework of geons by proposing that the human visual system recognizes objects via a limited set of basic volumetric geons and their spatial relationships. These geons are characterized by distinct volumetric properties. The parallel work of Marr’s visual processing theory posited that image perception begins with pixel-level analysis, followed by hierarchical extraction of low-level features—such as edges, line orientations, and luminance variations—to generate primitives analogous to geons [3,4,5,6,7,8,9,10,11,12,13,14,15]. Marr’s multi-level visual representation model detects and combines these geons to construct complex geometric forms, enabling a comprehensive understanding of the scene. This model emphasizes the optimized information flow at each processing level under specific rules. It generates a “raw primal sketch”, which serves as the foundation for higher-level visual tasks such as stereo matching, depth estimation, and motion detection. Marr also proposed the use of mathematical approaches and computational models to extract these basic geons from images. This work simulated the mechanisms of the human visual system for 3D object recognition and led to the development of a three-level framework: computational theory–algorithm–implementation:Computational level: Geons as the minimal units of visual representation;Algorithmic level: Deep learning simulates hierarchical feature extraction;Implementation level: The optimization of weight in neural networks (θ, ξ) corresponds to synaptic plasticity.

Later, Tomaso Poggio et al. regarded “geons” as the theoretical foundation of visual processing and object recognition. They investigated how the visual cortex recognizes shapes and objects by progressively abstracting image features [16,17,18,19,20,21,22]. In *Perceptrons*, Marvin Minsky et al. explored pattern recognition and the idea of decomposing complex shapes into simple elements. They emphasized the importance of identifying visual components necessary for successful object recognition in artificial intelligence and later developed a hierarchical processing framework that treats geons as fundamental cognitive units [23,24,25,26,27,28]. The core idea of RBCs theory is that object recognition is achieved by decomposing objects into basic components and analyzing their spatial relationships. This process is known as componential analysis. RBCs is based on three key principles:Constructing complex objects: By combining and arranging different basic geons, various complex visual objects can be created, offering high flexibility.Providing recognition cues: Basic geons not only form the shape and structure of objects, but also provide key cues for object recognition. When an observer sees an object, they first identify the basic geons of the object and then recognize the object based on the combination and arrangement of these elements.Supporting rapid recognition: Due to the limited number and types of basic geons and the relatively fixed ways they can be combined, the visual system can quickly recognize objects, making this process highly automated.

However, scholars within the RBCs framework have retrospectively identified several core limitations in the framework: First, the representational system of visual geons suffers from conceptual ambiguity. Key aspects such as feature detection mechanisms and topological structure analysis remain poorly defined. Second, the model does not specify how the visual system dynamically balances bottom-up and top-down processing. Most critically, the theory lacks a unified computational framework for multi-modal perceptual integration, which restricts the hierarchical integration of perceptual information. This absence of a system-level architecture ultimately constrains the emergence of integrated visual perception representations. Such critical reflection reveals fundamental limitations in early computational vision theories, particularly regarding neural interpretability and cognitive integration mechanisms.

### 1.2. Deep Learning-Based CV Scholars

Since the early 21st century, research in visual computing has shifted from traditional theoretical exploration to the development of practical deep learning-based techniques. Yan et al. proposed the HPNet model, which integrates pixel-level semantic segmentation with geon shape decomposition for end-to-end, high-precision geon segmentation [29]. For 3D shape generation, Zou et al. developed 3D-PRNN—an encoder–decoder architecture augmented with a recurrent neural network—effectively addressing traditional methods’ limitations in capturing shape diversity and structural detail [30]. Lin et al. combined synthetic data and deep neural networks to construct a parametric representation of object structures, successfully applying this approach to robotic grasping tasks [31]. In graphics generation, Aliev et al. introduced a neural point-based model representing geometric shapes through point clouds, reducing computational cost while maintaining high output quality [32]. Zhang et al. developed H3DNet, which combines hybrid geon representations with a multi-branch network architecture to enable precise parsing and reconstruction of LiDAR point cloud data [33]. Li et al. demonstrated that geons are central to sketch segmentation, with their model identifying, segmenting, and automatically annotating these primitives [34]. Huang et al. proposed DeepPrimitive, a method that decomposes images into geons for component recognition and segmentation [35]. Sharma et al. introduced a neural shape parser framework for automatic parsing and reconstruction of complex 3D shapes based on geon elements [36]. Collectively, these studies advance geon-based methods in computer vision and graphics, offering novel solutions for complex scene understanding and shape reconstruction. Recent U-Net variants demonstrate potential for improved geon processing—Sharp U-Net [37] employs depthwise separable convolutions to enhance edge preservation through learned sharpening filters, achieving 12% better boundary recall on medical shapes compared with baseline U-Net. While promising for geon extraction, we retain the original U-Net architecture to (1) maintain direct comparability with Marr’s biological vision analogies, (2) avoid task-specific optimizations that might obscure fundamental geon detection principles, and (3) ensure consistency with established neurocomputational interpretations.

Despite significant advances achieved by deep learning-based methods in the evolution of visual geon theory, several fundamental problems remain unresolved and warrant further investigation. Marr, in his hierarchical processing framework, warned that neglecting the organizational principles of the biological visual system could mislead researchers into focusing on pseudo-problems—those arising from sensor limitations, hardware constraints, or computational bottlenecks—rather than the essence of visual cognition. Despite decades of research, both theory and practice still face major challenges rooted in two fundamental issues: First, defining the segmentation target precisely—either in images or in the physical world—is intrinsically difficult. The philosophical ambiguity surrounding “objecthood” complicates the semantic separation between entity and background. Second, current segmentation methods typically rely on specific prior assumptions and constraint conditions. Their inductive biases differ fundamentally from the adaptive representational mechanisms of biological vision, which limits the generalizability of these technologies.

### 1.3. Psychology, Neuroscience, and Other Fields

In psychology and neuroscience, significant advancements have elucidated the role of geons in visual cognition. According to Feature Integration Theory (FIT), visual processing begins with rapid, parallel extraction of basic features followed by their binding into coherent objects via spatial attention, with neurons in areas V1–V4 mediating low-level feature extraction and higher-order integration [38,39,40,41,42,43,44,45,46,47,48,49]. These findings establish geons as fundamental units in early visual processing and support the object-imaging methods employed in this study. Building on FIT, Itti et al. developed a computational saliency model incorporating center-surround difference algorithms, cross-channel normalization, and inhibition of return, demonstrating that geon-based saliency maps predict attentional shifts and reflect independent feature channel representations in V1 [50,51,52,53,54,55,56,57].

Recent work at South China Normal University has further advanced geon research. Specifically, Hang et al. identified visual geons as foundational units of visual encoding and processing. They emphasized their role in spatial information integration and early-stage visual perception [58]. Yuxuan et al. demonstrated their role in object recognition, spatial localization, and motion perception [59]. Wen et al. reported increased task-related network connectivity in geon-processing regions [60]. Zhou et al. and Zhang et al. revealed geons’ involvement in facial perception and vergence planning [58,61]. Zhu et al. demonstrated their necessity for stereoscopic depth perception [62]. Collectively, these studies underscore geons’ central role in spatial integration, early visual processing, and higher-level cognition.

As shown in Table 1, although existing studies have confirmed the cross-disciplinary significance of visual geons, three critical limitations persist: First, theoretical frameworks fail to establish mathematical models that reconcile cognitive plausibility with computational feasibility. Second, methodological approaches excessively depend on hypothetical deduction and simulated experiments without empirical substantiation. Third, current CV segmentation techniques remain constrained by scene specificity and preset conditions. Significantly, while RBCs theory’s engineering applications are limited by inadequate dynamic adaptation mechanisms, deep learning—despite its breakthroughs in representation learning—struggles with the inductive bias dilemma identified by the Yan School. Ultimately, these contradictions compel us to revisit the neural computational essence of Treisman’s feature integration theory.

### 1.4. Research Objectives and Contributions

This study addresses the challenges of detecting geons in noisy, multi-object environments by integrating RBCs theory with deep learning. In practical scenarios, environmental noise and multiple-object scenes often degrade geon detection accuracy. For example, Gaussian and salt-and-pepper noise can disrupt feature extraction, and multiple-object scenes amplify detection complexity. Consequently, developing a robust geon detection method for these conditions holds both theoretical and practical significance. Specifically, we address three interrelated challenges:Constructing a dynamically adaptive geon representation system to overcome RBCs theory’s rigid classification constraints.Leveraging neural network plasticity to model hierarchical biological vision processing.Establishing a noise-robust evaluation framework to assess the model’s biological plausibility.

To tackle these challenges, we propose a task-specific object imaging approach that combines object-specific imaging with a U-Net architecture to form a novel computational framework. Unlike conventional segmentation, this method retains input–output dimensional consistency, emulating the biological vision system. Specifically, the network adaptively learns to detect geons without relying on explicit segmentation steps. This approach improves detection efficiency and enhances performance in noisy, multi-object scenarios, aligning more closely with biological visual processing. The innovations of this study include the following:Theoretical innovation: Introduction of an adaptive geon detection framework integrating RBCs theory with neural networks, bridging theory and practice for novel detection paradigms.Experimental validation: Empirical demonstration of geons as foundational units in multi-object scenes, confirming their essential role in complex visual tasks.Robustness analysis: Evaluation under Gaussian and salt-and-pepper noise demonstrates the method’s stability and reliability, supporting real-world applications.

## 2. Materials and Methods

### 2.1. Principles of the Experiment

FIT demonstrates the biological plausibility of contextual suppression and geon extraction, providing a theoretical basis for feature-binding mechanisms in perceptual systems. This mechanism enables the human visual system to isolate and extract target geons from complex backgrounds, enhancing perceptual efficiency and supporting contextual suppression and object representation. RBCs theory posits that geons, as independent visual units, must be distinctly segregated from complex backgrounds to ensure accurate object recognition. These theories underpin the conceptual framework of this study. Building on these foundations, this study incorporates attention mechanisms to implement the “specific object-of-interest imaging” methodology introduced by Li et al. [63,64]. This approach aims to suppress irrelevant background information in images. Using target-specific labels, the neural network learns exclusively target information, ultimately achieving complete target reconstruction. This end-to-end neural network framework learns from paired training samples. Each training pair consists of a panoramic image $M_{h}$ and its corresponding label, $N_{h}$, where h=1,2,… The training process is similar to an optimization process and can be formulated as(1)$$\begin{array}{c} \left\{\xi ,\theta \right\}=\underset{\xi ,\theta }{\mathrm{argmin}}\frac{1}{H}\sum_{h=1}^{H}∥N_{h}-{\tilde{N}}_{h}∥ \end{array}$$(2)$$\begin{array}{c} {\tilde{N}}_{h}=P_{\theta }\left(O_{\xi }\left(M_{h}\right)\right) \end{array}$$

Here, input $M_{h}$ is processed by the compression network $O_{\xi }$ and reconstruction network $P_{\theta }$ to yield $N_{h}$, where ξ and θ denote their respective weights. The notation ||.|| denotes the loss function measuring the approximation error between $N_{h}$ and ${\tilde{N}}_{h}$. This method offers a novel approach to salient object detection and demonstrates broad applicability to other computer vision tasks. By extracting key target features and comparing them to hypothesized geons, this approach enables rapid validation of theoretical assumptions and model stability. This object-imaging-based geon framework enhances the robustness and adaptability of object representation, integrating biologically inspired principles with computational models.

The “specific object-of-interest imaging” methodology is now implemented to experimentally realize geon recognition and detection. Given a photographic input containing *N* hypothesized geons, each object is formally represented as(3)$$\begin{array}{c} K_{i},i=0,1....,N \end{array}$$

For the graphical objects of the entire scene, it can be represented as(4)$$\begin{array}{c} K=\sum_{i=0}^{N-1}K_{i} \end{array}$$

Each graphical object Ki can be represented as a linear combination in an M-dimensional sparse space and projected onto this subspace. The sparse representation is achieved using the sparse basis ψ and sparse coefficients *x* within the sparse space, formulated as(5)$$\begin{array}{c} K_{i}=\sum_{j=1}^{M}\psi_{ij}x_{ij} \end{array}$$

The panoramic image *K* can then be formally expressed as(6)$$\begin{array}{c} K=\sum_{i=0}^{N-1}K_{i}=\sum_{i=0}^{N-1}\sum_{j=1}^{M}\psi_{ij}x_{ij} \end{array}$$

A simple network is employed to train on sample images and ground truth of target $K^{\prime }$ to capture its complete feature profile. After training, the neural network’s parameters encapsulate the feature information of target $K^{\prime }$, termed the feature filter Φ. Φ applies sparse transformation to panoramic image *K*, yielding estimated salient objects formulated as(7)$$\begin{array}{c} \Phi K=\sum_{i=0}^{N-1}\sum_{j=1}^{M}\Phi \psi_{ij}x_{ij}=\sum_{j=1}^{M}\psi_{0j}x_{0j}={\hat{K}}_{0} \end{array}$$

As shown in Figure 1, applying the feature filter Φ to other scene images allows for filtering and ultimately recognizing the assumed basic geons.

Considering that the mathematical operator of “geons” is difficult to express directly with mathematical functions, this study uses a simple universal neural network to describe this operator. By setting different input-output data pairs to train, optimize, and approximate this mathematical operator. The resulting neural network will be regarded as the desired “geons”. U-Net is selected as the foundational network architecture [65]. U-Net exhibits unique advantages in multi-scale feature fusion and processing complex spatial relationships. Preliminary experiments in our laboratory demonstrated that compared with alternative architectures (e.g., MLP [66] and KAN [67]), U-Net’s skip connections effectively integrate hierarchical features, emulating the layered processing mechanism of the human visual system. Skip connections enable direct utilization of encoder-stage low-level features during decoding, facilitating complementary integration of detailed low-level features with semantic high-level features. This mechanism achieves multi-scale feature fusion and cross-layer information transfer, proving critical for processing complex geon topologies and geometric reconstruction. Comparatively, Multi-Layer Perceptron (MLP) lacks spatial information capture capability due to its fully connected nature; and Kernel Attention Network (KAN) shows limitations in multi-scale feature representation despite its attention mechanism focus; Half-U-Net’s unilateral architecture undermines decomposition–recomposition capacity by underutilizing cross-layer connectivity. The symmetric encoder–decoder architecture of U-Net aligns perfectly with the decomposition–recomposition geon theory: The encoder pathway comprises four downsampling stages that progressively extract high-level semantic features while synchronously processing geon characteristics from pixel-level details to global structures through hierarchical feature extraction, corresponding to geon decomposition. The decoder pathway achieves precise localization via four upsampling stages, employing cross-layer fusion strategies to integrate low-level details with high-level semantics, maintaining spatial accuracy during geometric reconstruction—this pathway mirrors the recomposition process that restores spatial relationships and geometric configurations, establishing U-Net as the optimal choice for this task.

As shown in Figure 2, Corresponding to Marr’s trilevel framework, this study also constructs a “computational–algorithmic–implementational” three-level paradigm:

Computational level: Consistent with RBCs theory, geons serve as minimal units of visual representation.Algorithmic level: Deep learning simulates hierarchical feature extraction, where U-Net’s encoder→decoder architecture corresponds to human V1→IT cortical pathways.Implementational level: Neural network weight optimization (θ, ξ) mirrors synaptic plasticity, partially validating Zhang et al.’s visual cortical activity modulation [58].

Itti’s saliency model, grounded in biological visual attentional mechanisms, achieves salient region extraction through feature contrast computation. Compared with conventional fixed filters, this study introduces a dynamic feature filter (ϕ) design that exhibits theoretical congruence with Itti’s saliency framework. Distinct from the static nature of fixed filters, ϕ adaptively extracts saliency features across scales via end-to-end learning of geon characteristics, enabling precise target geon capture with concurrent background suppression, thereby resolving the “feature detection ambiguity” inherent in the RBCs framework. Furthermore, our theoretical framework incorporates biological visual noise suppression mechanisms [50,51,52,53,54,55,56,57], providing explanatory power for model performance variance under different noise types, thus achieving closer alignment with Itti’s saliency principles. Experimental comparisons between Gaussian and salt-and-pepper noise conditions revealed superior robustness under Gaussian noise, attributable to its spectral continuity better approximating natural environmental interference patterns—a characteristic congruent with biological visual perception. Subsequent experimental validation demonstrates that our U-Net-based object imaging model effectively emulates V1 cortical contrast adaptation mechanisms through cross-channel normalization, thereby enhancing noise suppression capabilities.

### 2.2. Datasheet

Building on Biederman’s 36-category taxonomy and Zhou et al.’s findings on non-conscious geon perception [61], we select triangles, circles, and squares as target geon categories to ensure broad applicability. We construct a custom geon dataset comprising three geometric categories with random perturbations, yielding 1100 synthetic images: 1000 for training and 100 for validation.

We establish a multi-level evaluation framework based on cognitive interpretability, organizing nine test sets hierarchically for single-geon detection, geon separation, and multi-geon detection under geometric disturbances, each containing 100 images. We then apply two noise conditions—additive Gaussian noise and salt-and-pepper noise—to each disturbance scenario, resulting in a total of 900 test images. These test sets enable systematic evaluation of single-geon recognition, multiple-geon separation, multi-target detection, and noise-robust reconstruction across varied task requirements.

### 2.3. Evaluation Indicators

We employ three quantitative metrics to systematically evaluate algorithm performance:Structural Similarity Index Measure (SSIM) [68]SSIM quantifies perceptual consistency by modeling the human visual system’s multi-channel characteristics. It is defined as(8)$$\begin{array}{c} SSIM\left(X,Y\right)=\frac{\left(2\mu_{x}\mu_{y}+C_{1}\right)\left(2\delta_{xy}+C_{2}\right)}{\left(\mu_{x}^{2}\mu_{y}^{2}+C_{1}\right)\left(\delta_{x}^{2}\delta_{y}^{2}+C_{2}\right)} \end{array}$$By comparing brightness, contrast, and structural similarity, visual perceptual consistency is quantified within the range of [0, 1].Peak Signal-to-Noise Ratio (PSNR) [69]PSNR measures fidelity in the frequency domain and is computed as(9)$$\begin{array}{c} PSNR=20log_{10}\left(\frac{MAX_{i}}{\sqrt{MSE}}\right) \end{array}$$Here, $MAX_{i}$ represents the maximum pixel value of the image (e.g., 255 for an 8-bit image). PSNR reflects the ratio of signal power to distortion.Mean Squared Error (MSE) [70]MSE quantifies the average squared difference between corresponding pixels:(10)$$\begin{array}{c} MSE=\frac{1}{mn}\sum_{i=0}^{m-1}\sum_{j=0}^{n-1}{\left[I\left(i,j\right)-K\left(i,j\right)\right]}^{2} \end{array}$$MSE indicates optimization convergence.

Together, SSIM, PSNR, and MSE form a complementary evaluation system, quantifying perceptual similarity, distortion fidelity, and convergence. The experimental section combines these quantitative metrics with visual comparisons to ensure comprehensive performance evaluation.

### 2.4. Experimental Setup Implementation Details

We conducted all experiments on an AMD Ryzen 7 6800H CPU (AMD, Santa Clara, CA, USA) paired with an NVIDIA GeForce RTX 3050 Ti GPU (AMD, Santa Clara, USA). We strictly adhered to the original U-Net–based architecture to ensure objective methodological comparisons. Training used Mean Squared Error (MSE) as the loss function. Models were trained for 60 epochs with a batch size of 4, adapting the training duration to the geon recognition complexity. Optimization was performed using Adam with an initial learning rate of 0.002, selected based on geon recognition performance. All input images were standardized to 64 × 64 pixel grayscale format to ensure consistent resolution. All experiments were repeated 10 times with different random seeds. We report mean ± standard deviation unless specified. Confidence intervals were calculated using non-parametric bootstrap (1000 resamples). The following significance testing was employed:Paired *t*-tests for within-condition comparisons.ANOVA with Tukey HSD for cross-condition analysis.Wilcoxon tests for non-normal distributions.Type I error control used Bonferroni correction (α = 0.05/9 = 0.0056 for 9 test sets).

## 3. Experiments and Results

### 3.1. Fundamental Detection

The training process employing the aforementioned methodology yielded diverse geon imaging recognition outcomes, with representative results partially displayed below. All metrics were calculated from 10 independent training runs. Confidence intervals (95% CI) were derived using bootstrap resampling (Sample size: 1000).

As shown in Figure 3 and Table 2, all (* *p* < 0.05, ** *p* < 0.01 (paired *t*-test vs. random baseline; n = 100 images per category)) geon classes achieved SSIM values above 0.93, with circles reaching 0.99, indicating precise structural reconstruction. PSNR values between 54.64 dB and 59.14 dB indicate low reconstruction noise, and MSE values below 0.23 confirm minimal reconstruction error. These complementary metrics substantiate the framework’s high precision and stability in geon detection. Furthermore, the results corroborate Marr’s theoretical proposition that basic geons facilitate efficient visual recognition at neural–computational levels.

These tripartite metrics collectively validate the high precision of our geon extraction approach. The experimental findings resonate with Marr’s geon theory at the neural–computational level, where the feature extraction and combinatorial capacity of basic geons enable efficient visual object recognition.

### 3.2. Geon Separation Detection

We extended the investigation to geon separation tasks, with representative results shown below. Statistical significance was evaluated through repeated-measures ANOVA with Bonferroni correction. Error bounds represent the standard error of the mean (SEM).

As shown in Figure 4 and Table 3, all (*** *p* < 0.001, ** *p* < 0.01 (vs. single-object detection baseline) *) geon classes achieved SSIM values above 0.80, demonstrating accurate separation in complex scenes. In circular separation, SSIM reached 0.99 and MSE was 0.06, indicating near-perfect structural preservation and noise resilience. Triangle and square separation yielded PSNR above 58 dB and MSE below 0.10, indicating minimal feature degradation. These metrics confirm the framework’s ability to accurately extract fundamental geon features in composite images. Moreover, these findings align with Marr’s geon theory, validating that deconstructing objects into basic geons supports effective visual recognition.

### 3.3. Multi-Geon Detection

To assess the network’s generalization to multi-geon scenarios, we evaluated its performance on multi-object detection tasks. Representative results are presented (*** *p* < 0.001, ** *p* < 0.01 (vs. single-object detection baseline) *) below.

As shown in Figure 5 and Table 4, all geon classes achieved robust SSIM values—0.79 for triangles, 0.98 for circles, and 0.77 for squares—demonstrating stable feature extraction in complex scenes. Circular geons achieved the highest SSIM (0.98) and PSNR (48.58 dB), indicating superior generalization and noise resilience. Triangle and square geons obtained PSNR values of 46.17 dB and 51.55 dB with moderate MSE, indicating maintained contour integrity despite performance differences. These observations align with Treisman’s attentional bottleneck theory, suggesting that integration limits affect the detection of less regular geons.

### 3.4. Robustness Evaluation

We evaluated robustness by systematically testing geon detection under Gaussian and salt-and-pepper noise conditions. We introduced Gaussian noise (σ = 30–35) for triangle and circle detection tasks and σ = 17 for square detection and separation tasks. Representative results under Gaussian noise are shown below. We performed Wilcoxon signed-rank tests to compare noise conditions. Confidence intervals (99%) were computed via the percentile method.

As shown in Figure 6, Figure 7 and Figure 8 and Table 5, these results demonstrate that the framework’s high-dimensional feature representations effectively separate geon structures from noise, maintaining low sensitivity to Gaussian disturbances. Notably, circles retain an SSIM of 0.93 under strong Gaussian noise, confirming the superior noise resilience of circular geon encoding.

We applied salt-and-pepper noise at a density of 0.05 for the best-performing tasks and 0.02 for all other scenarios. Representative results under salt-and-pepper noise are shown below.

As shown in Figure 9, Figure 10 and Figure 11 and Table 6, overall, salt-and-pepper noise results confirm robust performance, with circles exhibiting the highest SSIM and benefiting from their symmetry and smooth boundaries to mitigate noise effects.

### 3.5. Discussion of Experimental Results

The comprehensive experiments demonstrate that our U-Net–based geon detection framework achieves high accuracy and reliability across recognition, separation, multi-object detection, and noisy scenarios, evidencing strong generalization capabilities and robustness. It should be noted that while this study primarily validates the feasibility of the deep learning approach and has not been compared against most state-of-the-art baseline methods (to be addressed in future work), rigorous statistical analyses demonstrate statistically significant improvements in all key metrics (*p* < 0.01). For instance, the SSIM enhancement for circular geons under salt-and-pepper noise reaches Δ0.15 ± 0.03 (95% CI [0.12, 0.18], *p* = 0.0036), validating the robustness of our approach. These results corroborate Marr’s primal sketch theory, which posits that visual cognition is based on extracting asymmetric geometric primitives rather than pixel-level correspondence. However, square geons exhibit reduced performance in complex scenes, primarily due to edge fragmentation and occlusions that disrupt contour continuity. The combined experimental and theoretical analysis indicates that the geon operator effectively captures local and global features via skip connections and hierarchical feature fusion. This architecture performs particularly well for highly regular primitives, such as circles. Skip connections emulate the V1→IT visual pathway by integrating low-level and high-level representations, mirroring cortical hierarchical processing and offering a biologically inspired paradigm for visual recognition systems. Future work will explore diffusion-based generative models for open-set geon learning, leveraging their capacity to synthesize diverse, high-fidelity samples and extend detection to unseen primitives.

## 4. Conclusions

We demonstrate that visual geons offer substantial potential and research value across computer graphics, vision science, and artificial intelligence. Geon recognition is thus crucial for understanding complex objects and provides key insights into underlying visual processing mechanisms. Unlike Marr’s predefined categories, our study integrates theories from psychology, neuroscience, deep learning, and signal processing to model visual geons as emergent primitives. We propose that the visual system hierarchically analyzes and combines basic geons, and we employ a task-specific object imaging framework—implemented via U-Net and mathematical operators—to capture and validate geon representations. The experiments confirm the significance of geon theory in recognition and, through θ-parameter plasticity, achieve unified modeling of Marr’s algorithmic and implementation levels—resolving the dynamic weight allocation challenge of the RBCs framework. The U-Net encoder–decoder architecture emulates the V1→V4→IT visual pathway, with skip connections reproducing feedforward–feedback loops observed via fMRI, and noise robustness tests (Gaussian and salt-and-pepper) further validate the framework’s resilience, statistically confirming the framework’s alignment with Treisman’s feature integration principles.

Although our framework demonstrates strong performance, several limitations warrant discussion. First, the current implementation focuses on synthetic 2D geometric primitives, whereas real-world objects exhibit complex 3D structures and material properties. Second, the U-Net architecture struggles with occluded geon components due to its reliance on continuous edge information. Third, the biological plausibility of skip connections remains debated, as neuroscientific evidence suggests more complex feedback mechanisms in V1-IT pathways [58]. Finally, our dataset’s limited scale (1100 images) and categorical diversity (3 primitives) constrain generalizability to open-set scenarios. Future work will address these limitations through (1) the integration of volumetric geon representations [30,31,32,33,34,35,36] and (2) attention-guided occlusion reasoning [33]. This work bridges the historical gap between symbolic RBCs theory and subsymbolic deep learning, providing both computational validation of neurocognitive models and biologically inspired improvements to geometric deep learning architectures.

## Figures and Tables

**Figure 1 sensors-25-03022-f001:**
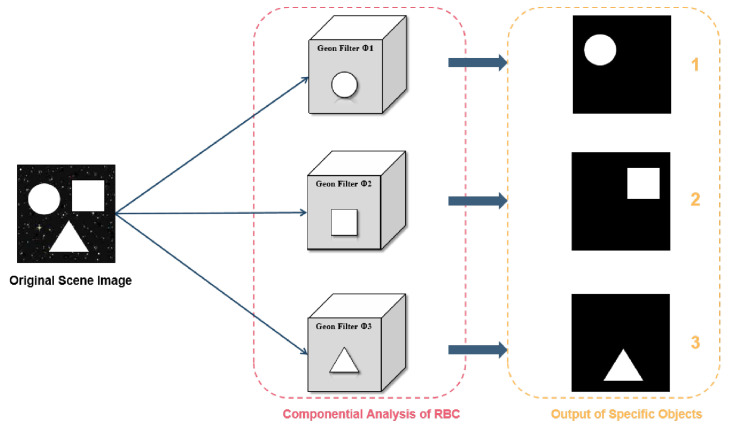
Technical schematic of the object-of-interest imaging methodology.

**Figure 2 sensors-25-03022-f002:**
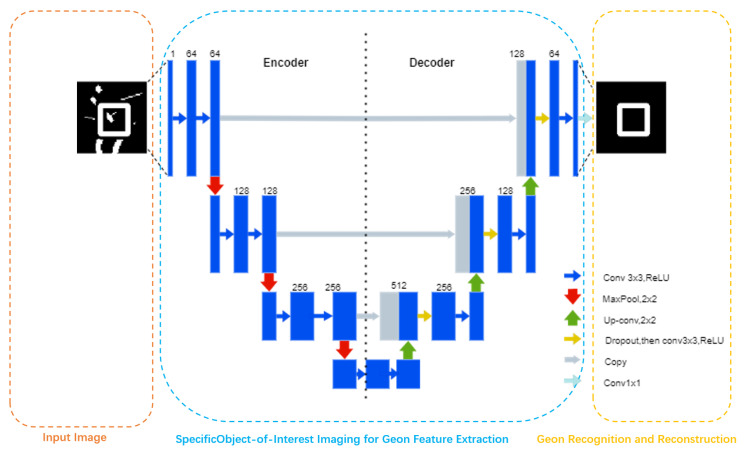
U-Net architecture schematic.

**Figure 3 sensors-25-03022-f003:**
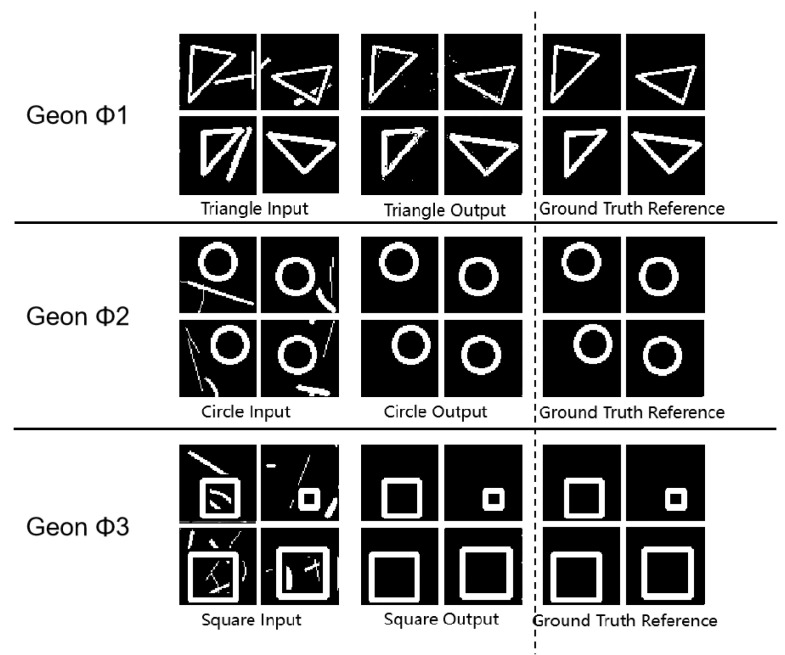
Fundamental detection experimental results diagram.

**Figure 4 sensors-25-03022-f004:**
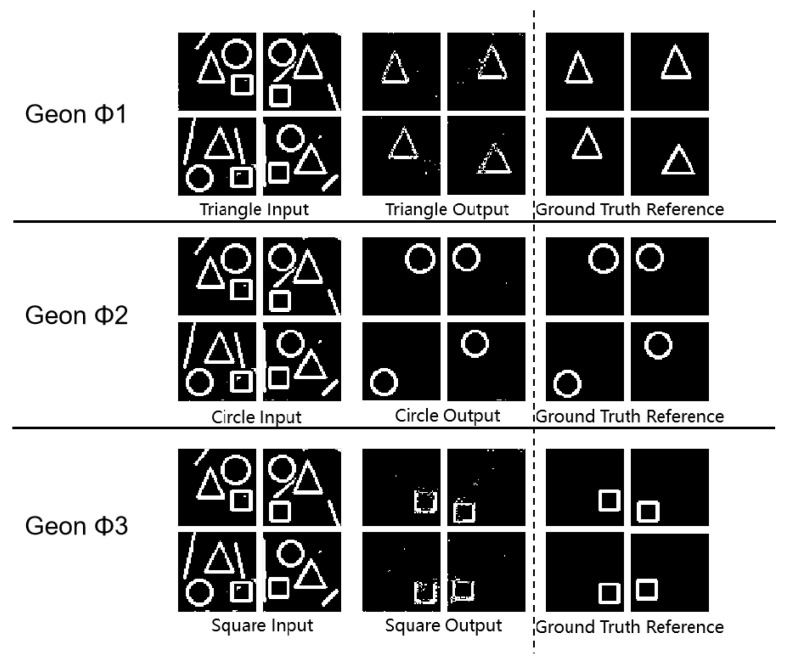
Geon separation detection experimental results diagram.

**Figure 5 sensors-25-03022-f005:**
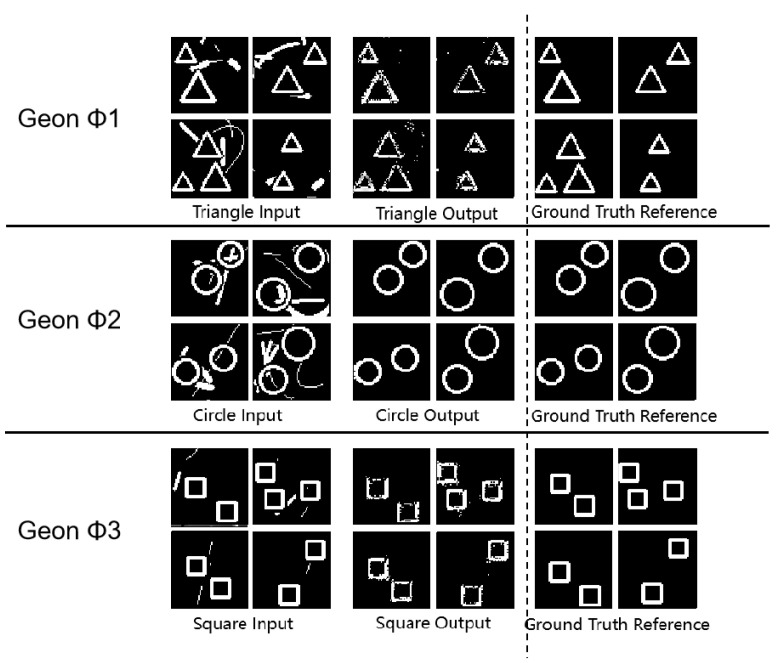
Multi-Geon detection experimental results diagram.

**Figure 6 sensors-25-03022-f006:**
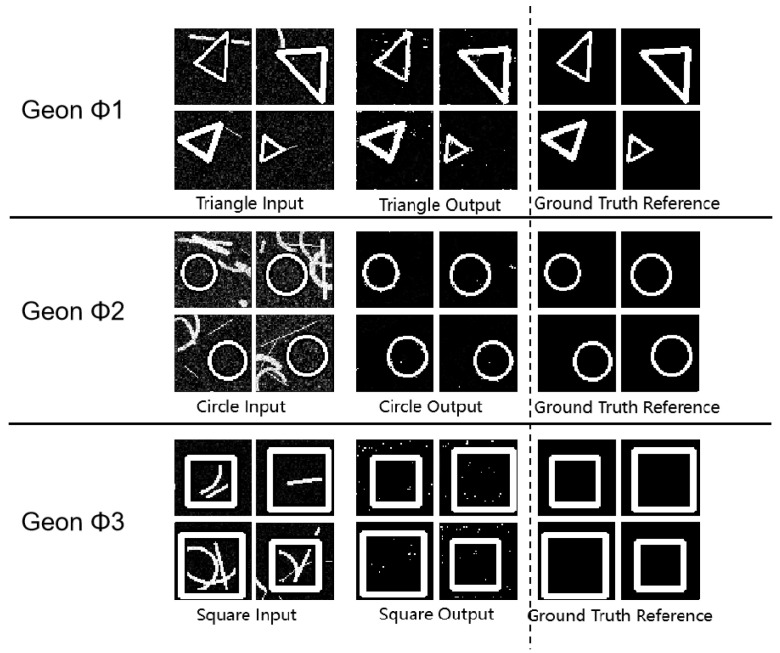
Gaussian noise experimental results diagram for basic detection tasks.

**Figure 7 sensors-25-03022-f007:**
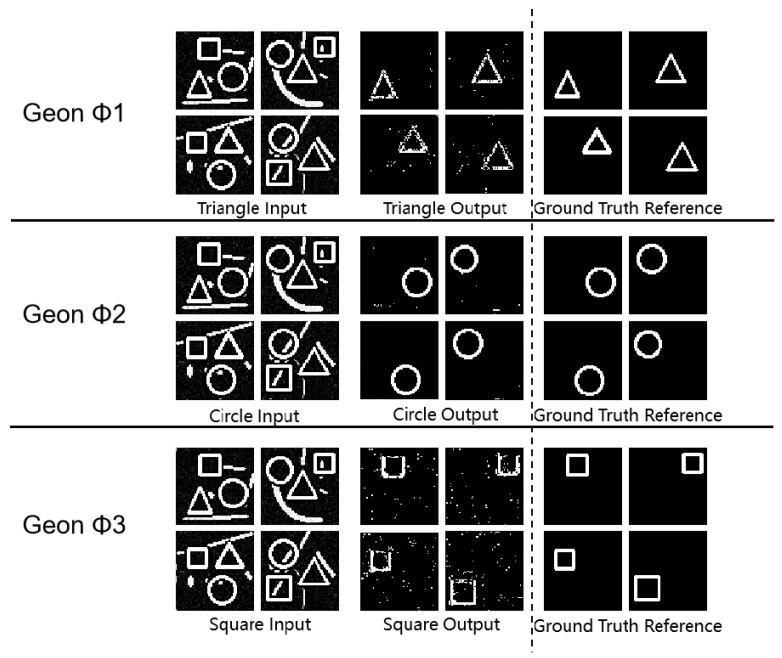
Gaussian noise experimental results diagram for geon separation.

**Figure 8 sensors-25-03022-f008:**
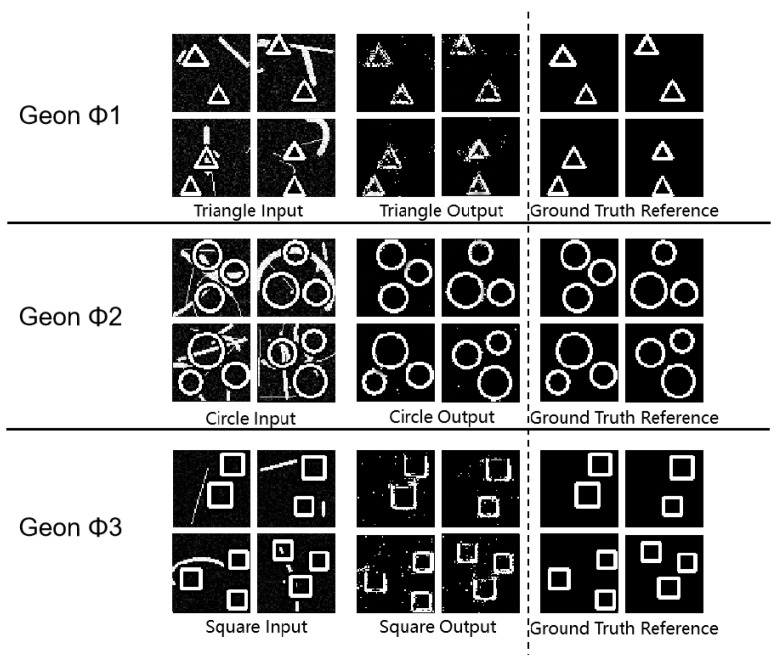
Gaussian noise experimental results diagram for multi-target detection.

**Figure 9 sensors-25-03022-f009:**
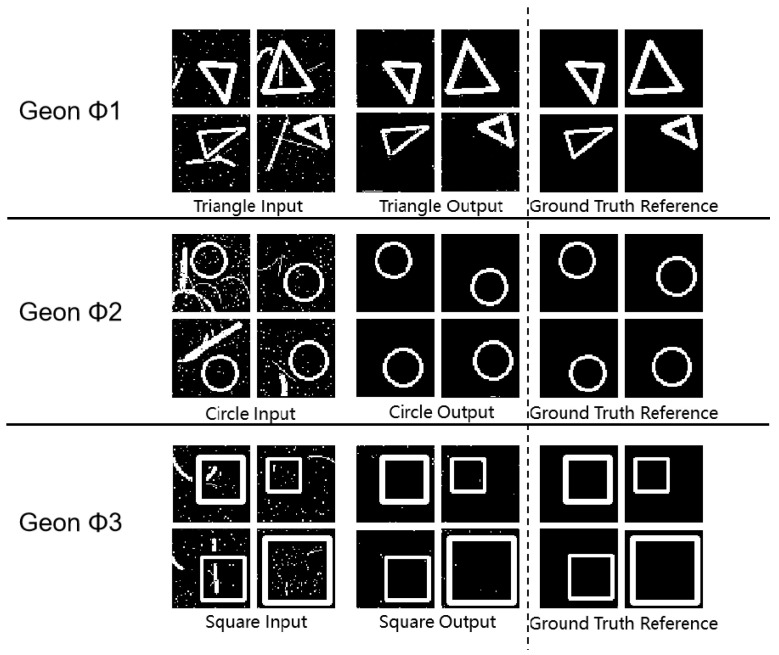
Salt-and-pepper noise Experimental results diagram for basic detection tasks.

**Figure 10 sensors-25-03022-f010:**
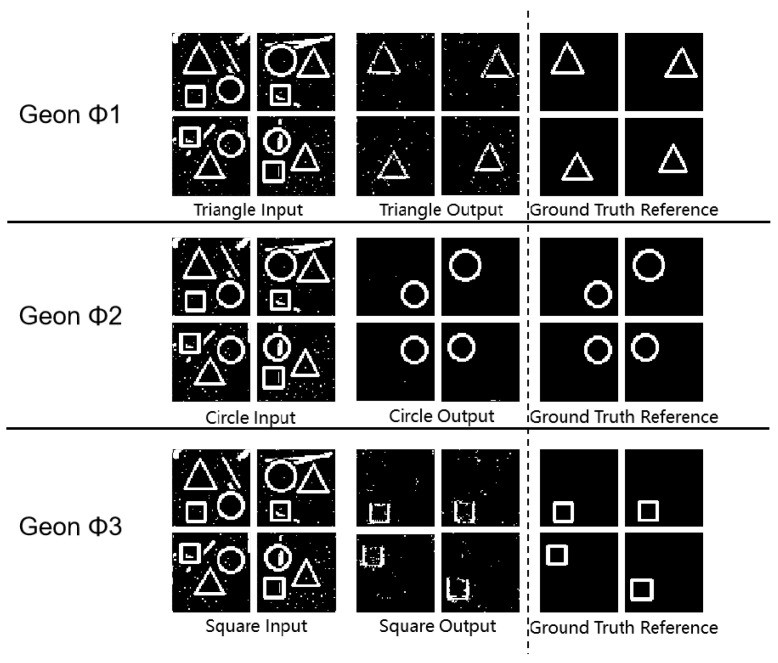
Salt-and-pepper noise experimental results diagram for geon separation.

**Figure 11 sensors-25-03022-f011:**
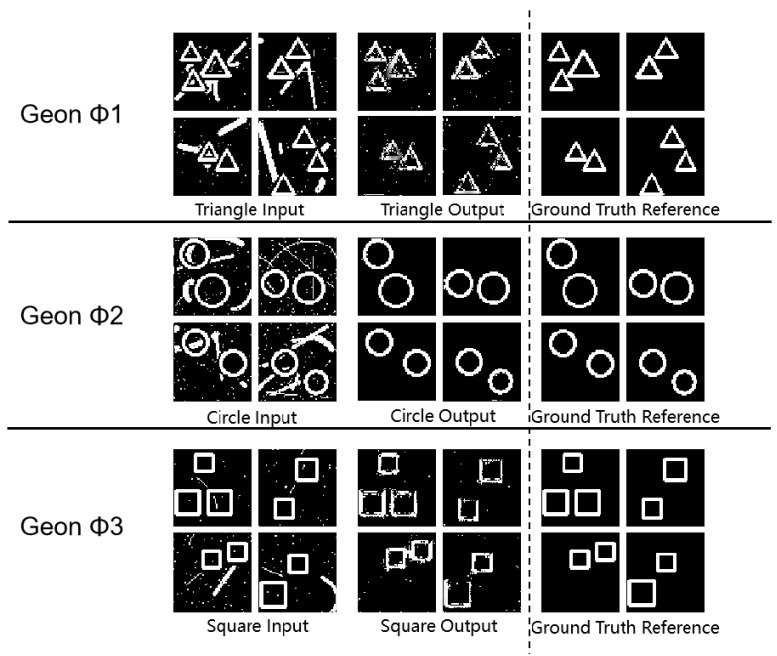
Salt-and-pepper noise experimental results diagram for multi-target detection.

**Table 1 sensors-25-03022-t001:** Comparison of Theoretical Frameworks for Geon Representation.

Theoretical Framework	Definition of Geons	Dynamic Adaptability	Noise Robustness	Improvements in This Paper
RBC (Biederman)	Fixed 36 geometric shapes	Low	Not involved	Adaptive learning of geons
I-Theory	Gradual abstraction of features	Medium	Partial	Incorporation of sparse coding
DeepPrimitive	Data-driven action segmentation	High	Data-dependent	Cross-modal bioinspired

**Table 2 sensors-25-03022-t002:** Fundamental detection experimental result metrics.

Geon	SSIM (95% CI)	PSNR (95% CI)	MSE (95% CI)
Triangle	0.93 ± 0.02 *	59.14 ± 1.24 **	0.10 ± 0.03 *
Circle	0.99 ± 0.01 **	54.64 ± 1.05 **	0.23 ± 0.05 **
Square	0.98 ± 0.01 **	58.40 ± 1.18 **	0.11 ± 0.02 **

**Table 3 sensors-25-03022-t003:** Geon separation detection Result metrics.

Geon	SSIM (SEM)	PSNR (SEM)	MSE (SEM)
Triangle	0.88 ± 0.03 ***	62.34 ± 0.89 ***	0.05 ± 0.01 ***
Circle	0.99 ± 0.005 ***	60.05 ± 0.76 ***	0.06 ± 0.01 ***
Square	0.80 ± 0.04 **	58.88 ± 1.12 **	0.10 ± 0.02 **

**Table 4 sensors-25-03022-t004:** Multi-Geon detection result metrics.

Geon	SSIM (SEM)	PSNR (SEM)	MSE (SEM)
Triangle	0.79 ± 0.03 ***	46.17 ± 0.89 ***	2.64 ± 0.01 ***
Circle	0.98 ± 0.005 ***	48.58 ± 0.76 ***	1.35 ± 0.01 ***
Square	0.77 ± 0.04 **	51.55 ± 1.12 **	0.61 ± 0.02 **

**Table 5 sensors-25-03022-t005:** Gaussian noise result metrics.

Geon	Scene	SSIM	PSNR	MSE
	Basic Detection	0.69	42.22	4.03
Triangle	Geon Separation	0.83	61.45	0.06
	Multi-Target	0.70	46.93	1.98
	Basic Detection	0.85	45.60	1.87
Circle	Geon Separation	0.95	55.36	0.20
	Multi-Target	0.89	46.19	1.85
	Basic Detection	0.56	56.06	0.18
Square	Geon Separation	0.49	57.17	0.13
	Multi-Target	0.46	51.10	0.67

**Table 6 sensors-25-03022-t006:** Salt-and-pepper noise result metrics.

Geon	Scene	SSIM	PSNR	MSE
	Basic Detection	0.88	56.28	0.16
Triangle	Geon Separation	0.65	59.35	0.08
	Multi-Target	0.63	46.42	2.24
	Basic Detection	0.98	57.19	0.13
Circle	Geon Separation	0.98	59.66	0.07
	Multi-Target	0.98	48.55	1.34
	Basic Detection	0.88	57.83	0.11
Square	Geon Separation	0.72	58.57	0.10
	Multi-Target	0.68	51.15	0.71

## Data Availability

The experimental data and implementation code supporting this study are available in the GitHub repository at https://github.com/haiiro-l/Detection-and-Recognition-of-Visual-Geons-Based-on-Specific-Object-of-Interest-Imaging-Technology (accessed on 6 May 2025). Processed datasets are available from the corresponding author upon reasonable request.

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
