# Peer review of "Detection and Recognition of Visual Geons Based on Specific Object-of-Interest Imaging Technology"

_sensors, 2025, doi:10.3390/s25103022_

Round 1

Reviewer 1 Report

Comments and Suggestions for Authors

The authors have explored the detection and recognition of visual geons by integrating classical visual theories with modern deep learning approaches, and have proposed an innovative "specific object of interest imaging" methodology. The attempt to simulate the hierarchical processing mechanism of the human visual system using the U-Net architecture demonstrates both theoretical insight and potential for practical application. However, several issues should be addressed before the manuscript can be considered for publication:

  1. The introduction enumerates a large number of theories, each discussed at considerable length, yet their direct relevance to the present study remains insufficiently articulated. The introduction spans over 170 lines, accounting for more than 40% of the total manuscript length. Given that this is a research article rather than a review, it is recommended that the introduction be substantially condensed and structurally optimized.
  2. The research objectives are not clearly delineated. Currently, the aims and innovations are embedded within the literature review, making them difficult to identify. It is suggested that a dedicated paragraph be added at the end of the introduction to clearly state the specific problem addressed, the proposed methodology, and the key contributions of this work.
  3. Sections 3.1 "Datasheet" and 3.2 "Evaluation Indicators" should be placed under Section 2 "Materials and Methods" instead of Section 3. Since Section 3 appears to be the conclusion, the current organization is inappropriate and requires adjustment.
  4. Statements regarding the model’s “robustness” and “geometric understanding capabilities” are somewhat subjective. A more thorough quantitative analysis of relevant parameters is recommended to substantiate these claims.
  5. The manuscript lacks a complete and structured Discussion section. Such a section is essential to interpret the results in depth, acknowledge the limitations of the method, and bridge the experimental findings with the research objectives and theoretical contributions.
  6. The formatting of references is inconsistent. The authors are advised to carefully check the journal’s guidelines for reference formatting and ensure that all citations follow a uniform and correct style.
Comments on the Quality of English Language

The English could be improved to more clearly express the research.

Reviewer 2 Report

Comments and Suggestions for Authors

The manuscript "Detection and Recognition of Visual Geons Based on Specific Object of Interest Imaging Technology" presents an interdisciplinary approach integrating deep learning. The proposal employs a U-Net-based architecture aligned with Marr and Biederman's theories and evaluates its robustness against noise conditions. I suggest the following improvements:

- Although existing theories adequately frame the paper, it would be desirable to make the novelty and concrete contribution of the proposed approach more explicit, both in the Introduction and in the Conclusions. For example, how does this proposal specifically differ from DeepPrimitive or HPNet beyond using the "object of interest imaging" technique?

- The abstract should present a more precise formulation of the problem, highlighting the methodological innovation and explicitly indicating the main quantitative results concisely. Likewise, the introduction should further motivate the need for improvement that prompted the work.

- The “Materials and Methods” section should include a complementary visual schematic or diagram of the mathematical formulations and a detailed justification of the choice of U-Net over alternative architectures for the task, considering their respective strengths and weaknesses.

- It is necessary to compare the proposal with other baseline methods on the same data how the proposal's advantage empirically.

- Additional statistical analyses should be incorporated, such as confidence intervals and p-values.

- Some technical terms, such as "geon separation," or "feature filterΦ" should be explained more clearly when introduced.

 - Figures are referenced but not discussed in depth.

Comments on the Quality of English Language

Overall, the manuscript is understandable, but some parts would benefit from grammatical revision to improve fluency and clarity. Consider professional English editing.

Reviewer 3 Report

Comments and Suggestions for Authors

The paper discusses an approach based on deep learning models to the detection and recognition of universal elements in images that allow us to describe objects of interest. The research results presented in it are of fundamental importance for developing an understanding of the mechanisms of perception of visual information. There are no significant comments on the content. As a recommendation, you can only indicate a desire to demonstrate the effectiveness of the proposed approach to analyzing real images. I believe that the paper can be recommended for publication in its present form or based on this recommendation.

Reviewer 4 Report

Comments and Suggestions for Authors

The work presented in this paper is on detection and recognition of visual Geons based on specific object of interest imaging technology. A great work indeed. Here are my observations and comments:

Abstract: It will be more appropriate to include the datasets, the evaluation metrics, the numerical results and the implications of the results in the abstract.

Introduction: The contributions of the work to knowledge should be added.

Related Work: This part is needed for better understanding of the existing techniques, their solutions and limitations.

Materials and Methods: Brief description should accompany the illustration of all the Figures and Tables, for better understanding.

Experiment and Results: Brief description should accompany the illustration of all the Figures and Tables, for better understanding. Moreover, existing results from recent methods should be compared to the results obtained by the proposed work, for better validation.

Conclusion: Supported by the results

References: The cited references are too old but relevant and appropriate.

Reviewer 5 Report

Comments and Suggestions for Authors

Title: 

Detection and Recognition of Visual Geons Based on SpecificObject of Interest Imaging Technology The article submitted  to MDPI (Sensor) deals with Geons types object detection and recognition. The research have the experimental set up to detect and recognize the object (cylinders, bricks, wedges, cones, circles and rectangles) in the visual worlds.  Merits: The problem is quite interesting and applicable to many domain of geographical and physical worlds. The representation of paper is good.  Demerits: 1) As the research topic revolves around computer vision problems detection and recognition.    2) The authors mentioned to remove the irrelevant background from the visual object. The quest is how it can be defined the irrelevant. What is level of optimization while the particular features of the object is affect by illumination conditions, occlusion etc. Give the mechanism to propose the paradigm for the shape analysis of the object.  3) have the authors gone through any kind shape feature described in CV to analyze the proper method of recognition and detection of the visual objects in complex environmental conditions.  4)  How the degree of compression and reconstruction of the objects assured if the object correctly recognized. Give he Mathematical formula to judge the detection and exact recognition of the object.  5) The measures of performance of the proposal used : PSNR, SSIM and MSE how these measure mathematically connect/affect the sparse bases and rank of the matrix of object features used to calculate the perception factor, filed of view of the objects.       

Round 2

Reviewer 1 Report

Comments and Suggestions for Authors

Accept in present form

Author Response

Dear Esteemed Reviewer,

We are sincerely grateful for your correspondence and for accepting our manuscript titled "Detection and Recognition of Visual Geons Based on Specific Object of Interest Imaging Technology" (ID: sensors-3548476). It is both an honor and a privilege to receive your constructive feedback and approval.

Your insightful critiques have been instrumental in refining the scientific rigor and clarity of our work. We particularly appreciate your suggestions, which have significantly elevated the manuscript’s contribution to the field. The editorial team’s meticulous guidance throughout the review process has also been invaluable.

In preparation for final submission, we will meticulously address all remaining editorial requirements and polish the manuscript further. Specifically, we plan to:

1.Revise the text to ensure absolute clarity, coherence, and conciseness, incorporating language edits as recommended.

2.Verify technical details, including equations, figures, and experimental data, to guarantee paper quality.

We kindly request your guidance should any minor adjustments still be required prior to publication. Please rest assured that we will prioritize addressing all final recommendations with utmost diligence.

Once again, thank you for your time, expertise, and patience. We are deeply indebted to the peer review process for transforming our work into a more impactful and scholarly publication.

Respectfully submitted,

Yonghao Wu ,Minyi Liu ,Jun Li* (Corresponding Author)

School of Electronic Science and Engineering, South China Normal University

Foshan, Guangdong 528000, China

Email: lizhjcd@scnu.edu.cn

Reviewer 2 Report

Comments and Suggestions for Authors

Most of the observations were addressed correctly, especially clarifying the contribution, the methodological justification, and the explanation of terms and figures.

I request that confidence intervals and p-values ​​be explicitly included, as indicated. I agree to leave the comparison with baseline methods as a future limitation.

The manuscript has improved considerably.

Author Response

Dear Esteemed Reviewer,

We are sincerely grateful for your correspondence and for accepting our manuscript titled "Detection and Recognition of Visual Geons Based on Specific Object of Interest Imaging Technology" (ID: sensors-3548476). It is both an honor and a privilege to receive your constructive feedback and approval.

Your insightful critiques have been instrumental in refining the scientific rigor and clarity of our work. We particularly appreciate your suggestions, which have significantly elevated the manuscript’s contribution to the field. The editorial team’s meticulous guidance throughout the review process has also been invaluable.

As for the problems you pointed out, we have corrected them in time after receiving the comments. Please review the latest manuscript of the paper.In preparation for final submission, we will meticulously address all remaining editorial requirements and polish the manuscript further. We plan to:

1.Revise the text to ensure absolute clarity, coherence, and conciseness, incorporating language edits as recommended.

2.Verify technical details, including equations, figures, and experimental data, to guarantee paper quality.

We kindly request your guidance should any minor adjustments still be required prior to publication. Please rest assured that we will prioritize addressing all final recommendations with utmost diligence.

Once again, thank you for your time, expertise, and patience. We are deeply indebted to the peer review process for transforming our work into a more impactful and scholarly publication.

Respectfully submitted,

Yonghao Wu ,Minyi Liu ,Jun Li* (Corresponding Author)

School of Electronic Science and Engineering, South China Normal University

Foshan, Guangdong 528000, China

Email: lizhjcd@scnu.edu.cn

Reviewer 5 Report

Comments and Suggestions for Authors

My comments are addressed ! 

Comments on the Quality of English Language

English is fine!

Author Response

(The authors gave the same response as above.)
